# Formation of Prenylated Chalcone Xanthohumol Cocrystals: Single Crystal X-ray Diffraction, Vibrational Spectroscopic Study Coupled with Multivariate Analysis

**DOI:** 10.3390/molecules24234245

**Published:** 2019-11-21

**Authors:** Iwona Budziak, Marta Arczewska, Daniel M. Kamiński

**Affiliations:** 1Department of Chemistry, University of Life Sciences in Lublin, Akademicka 15, 20-950 Lublin, Poland; iwona.budziak@up.lublin.pl; 2Department of Biophysics, University of Life Sciences in Lublin, Akademicka 13, 20-950 Lublin, Poland; 3Department of General and Coordination Chemistry and Crystallography, Institute of Chemical Sciences, Maria Curie-Sklodowska University in Lublin, pl. Marii Curie-Skłodowskiej 2, 20-031 Lublin, Poland

**Keywords:** prenylated chalcones, xanthohumol, crystal engineering, XRD, FTIR, Raman spectroscopy, principal component analysis (PCA)

## Abstract

Four novel xanthohumol (XN) cocrystals with pharmaceutically acceptable coformers, such as nicotinamide (NIC), glutarimide (GA), acetamide (AC), and caffeine (CF) in the 1:1 stoichiometry were obtained by the slow evaporation solution growth technique. The structure of the cocrystals was determined by single crystal X-ray diffraction. The analysis of packing and interactions in the crystal lattice revealed that molecules in the target cocrystals were packed into almost flat layers, formed by the O–H⋅⋅⋅O, O–H⋅⋅⋅N, and N–H⋅⋅⋅O-type contacts between the xanthohumol and coformer molecules. The results provided details about synthons responsible for crystal net stabilization and all hydrogen bonds observed in the crystal lattice. The main synthon was formed via the hydrogen bond between the hydroxyl group in the B ring of XN and coformers. The three-dimensional crystal lattice was stabilized by the hydrogen XN−XN interactions whereas the π–π stacking interactions played an additional role in layer binding, with the exception of low quality cocrystals formed with caffeine. Application of FTIR and Raman spectroscopy confirmed that the crystalline phase of obtained cocrystals was not a simple combination of individual components and completely different crystal phases resulted from the effect of intermolecular interactions. The multivariate analysis showed the changes in the spectra, and this technique can be applied in a combination with vibrational spectroscopy for fast screening of new crystal phases. Additionally, the solubility studies of pure XN and its cocrystals exhibited a 2.6-fold enhancement in XN solubility in aqueous solution for XN–AC and, to a lesser extent, for other cocrystals.

## 1. Introduction

Xanthohumol (2’,4’,4-trihydroxy-6’-methoxy-3’-prenylchalcone, XN, Figure 1) is a prenylated chalcone that occurs mainly in hops plant (*Humulus lupulus* L.), being the principal prenyloflavonoid of the female inflorescences [1,2]. Although the molecular structure of XN was identified in the 1950s [3,4], a growing interest in this molecule was observed only recently due to its numerous health-promoting properties [5,6]. XN has displayed a broad bioactivity spectrum, among other anticancer and chemoprotection [5,7], antioxidative [7,8], anti-inflammatory [9,10], and antiangiogenic activities [11,12] and even inhibition of osteoporosis in post-menopausal women activities [13] has been reported. 

Structurally, XN belongs to the open-chain flavonoids and is composed of two aromatic rings (A- and B-rings) connected by α, β-unsaturated ketone moiety and carrying a 3’-isoprenyl unit (3,3-dimethylallyl) [14] (Figure 1a). The presence of a prenyl moiety and hydroxyl groups in the XN structure determines largely its pharmacological activities [15,16,17]. The mechanism of covalent binding of XN with cytosolic proteins via electrophilic sites (e.g., the *α,β*-unsaturated carbonyl group) also strongly influences the distribution of XN and could be responsible for its unique health-oriented effects [18].

Unfortunately, the bioavailability of orally administrated xanthohumol is considered to be extremely low as a result of rapid excretion, or extensive metabolism [19]. Therefore, dietary consumption seems to be insufficient to obtain health related benefits. According to the biopharmaceutics classification system (BCS) the solubility and permeability properties are the major factors used to describe oral absorption of poorly water-soluble active compounds [20]. In this context, improvement of the XN bioavailability as well as its pharmacokinetic profile can depend to a great extent on the modification of physicochemical properties by creating binary-component systems in the form of cocrystals. As a new approach in the pharmaceutical industry, cocrystallization appears to be a promising way to modify water solubility, dissolution rate, and other physicochemical properties of biologically active compounds which contributes to their therapeutic effects [21,22,23,24]. Contrary to the forms of amorphous pharmaceutical substances, cocrystals have better defined physicochemical parameters and are thermodynamically more stable which is important during production and storage [25]. Cocrystals are single phase materials composed of two or more different molecular and/or ionic compounds generally in a stoichiometric ratio, bound by noncovalent interactions, predominantly hydrogen bonds, which are neither solvates nor simple salts [26]. As compared to the neat active pharmaceutical ingredient (API), cocrystals frequently display unique properties, which can be tailored through the changes of the coformer or packing of the molecules in the crystal lattice (cocrystal polymorphism) [27]. 

Polyphenols and flavonoids with the availability of plenty of hydrogen bond donors/acceptors are suggested to be worth of study from a crystal engineering perspective. Although in recent years the amount of studies on the cocrystallization of flavonoids in the aspect of their bioavailability has increased, there are few scientific reports on cocrystals formation by naturally occurring chalcones. In one of them, the enhanced dissolution rates for isoliquiritigenin (ISL) with the nicotinamide and isonicotinamide cocrystals were demonstrated, consequently aiming at improvement of ISL in vivo bioavailability [28]. 

The research of cocrystal structure and applications showed an exponential increase in the past decade, evident in the number of cocrystal structures deposited in the Cambridge Structural Database (CSD) and cocrystal related patent applications. Moreover, the polyphenols are more likely to form supramolecular synthons with such functional groups as carbonyl and aromatic nitrogen. In cocrystallization between the flavonoids and the compounds with organic nitrogen, the O−H···N heterosynthon has been reported to be the most competitive synthon [29,30,31,32]. 

Various techniques of cocrystal preparation have been reported such as solvent evaporation, solid state grinding, solution crystallization, slurry conversion, melt crystallization, hot melt extrusion, and spray crystallization [24]. The most commonly used technique for preparing cocrystals is the solvent evaporation method—therefore in this paper the cocrystallization of XN via the mentioned approach was exploited [33]. 

In light of the above-mentioned facts, the cocrystallization of xanthohumol with the commonly used coformers, nicotinamide (NIC), glutarimide (GA), acetamide (AC), and caffeine (CF) is reported herein (Scheme 1). The main concept behind this paper is to determine the structure of target cocrystals and indicate similarities and differences using various techniques such as single crystal X-ray diffraction, Fourier transform infrared (FTIR), and Raman spectroscopy as well as two multivariate analysis methods (i.e., principal component analysis (PCA) and cluster analysis (CA)). According to the current knowledge, the subject matter is innovative since so far cocrystallization has not been used to improve xanthohumol solubility. FTIR and Raman spectroscopy coupled with the multivariate statistical approach are effective methods to confirm new forms of cocrystals. It is worth noting that up to now there has existed only one crystallographic structure of XN present in the Cambridge Crystallographic Data Centre (CCDC) base [34]. 

## 2. Results and Discussion

The morphology of the selected cocrystals revealed that they are fragile and have defined facets. All of them have a layered structure and can be easily cleaved perpendicular to the *c* crystallographic direction (Figure 2a–e). Moreover, the XN–CF cocrystals have many defects that affect the quality of their structure (Figure 2e). Crystallographic parameters of the target crystals are listed in Table 1.

### 2.1. Single Crystal X-ray Diffraction

#### 2.1.1. XN Molecule in Cocrystals

The A ring and the α, β-unsaturated carbonyl group adopt almost a planar conformation in all cocrystal structures due to the intramolecular hydrogen bond O3A−H3A⋅⋅⋅O2A. The B ring is rotated with a dihedral angle of 1–10° with regard to the double bond plane. A planar conformation of the B ring is related to the intermolecular interactions in the crystal net. Rotations of the prenyl group around the C16−C12 bond are minor in all structures (see Figure 3). This is evident because the prenyl group interacts weakly with the surrounding atoms as it is visible in the large displacement ellipsoids of its atoms. In all cases the prenyl groups fill the gaps between planar parts in the XN molecules in the crystal lattices.

The most characteristic hydrogen bond motif in the structures of XN−NIC, XN−GA, and XN−AC is between the O4−H4a hydroxyl group and the O2 carbonyl oxygen originating from the neighboring XN molecule (Figure 4). This is the strongest hydrogen bond (1.64–1.92 Å) between XN molecules observed in these cocrystals and it is responsible for the XN layers’ stabilization in all cases. This hydrogen bond is the main building block of infinite chain C(8) (graph set notation). The structure of XN−CF in this case is an exception because the O2, O3, and O4 oxygen atoms (A ring) form hydrogen bonds with caffeine. This molecular arrangement leads to the rather weak XN interlayer hydrogen bond interactions as well as the π–π stacking resulting in the crystal cracking, and a low quality of the XN−CF structure (Table 1). In all cases due to a steric hindrance, the O5 does not participate in any intermolecular hydrogen bonds. The hydroxyl group O3H3 takes part in the intramolecular hydrogen bond to the O2 making XN molecular fragment more rigid and flat with all the OH moieties located on the plane (Figure 4). 

#### 2.1.2. Host-Coformer Interactions

The O1−H1a hydroxyl group participates in synthon formation with coformers in all cases (Figure 5). This hydrogen bond provides the main driving force for cocrystallization in all presented structures. Moreover, only the O–H⋅⋅⋅O, O–H⋅⋅⋅N, and N–H⋅⋅⋅O hydrogen-bonded interactions observed between the XN and coformer molecules were taken into account. The repeated heteromolecular motifs are presented in Figure 5.

In the case of XN−NIC, the XN molecule forms a synthon with carbonyl oxygen from nicotinamide molecule (1.997 Å) and nitrogen from the heterocyclic ring (1.925 Å) (Figure 5a). In the case of glutarimide the para hydroxyl group from the B ring forms two hydrogen bonds with oxygen (2.000 Å) and the amine group (3.855 Å) from the glutarimide molecule (Figure 5b). The small acetamide molecule adopts the optimal orientation to form two hydrogen bonds: the strongest one through carboxyl oxygen with the hydroxyl group from XN (1.692 Å) and weaker one with the O2 from the second neighboring XN molecule via the nitrogen atom (2.122 Å) (Figure 5c). Also, in XN−CF the main synthon is formed through the hydrogen bond from the para hydroxyl group (O1−H1a) to the carbonyl oxygen from caffeine (1.903 Å). In the structures XN−NIC, XN−GA, and XN−AC, the coformer molecules form infinite chains of hydrogen bonds in and out of the plane of XN layers (Figure 6, Figure 7, Figure 8 and Figure 9).

#### 2.1.3. XN−NIC (1:1)

The asymmetric unit comprises one XN molecule and one nicotinamide molecule which is in the center of symmetry. In the crystal packing nicotinamide molecules form a columnar assembly in the *c* direction which passes through the layers of XN molecules (Figure 6a). The distance between the XN layers is ~3.55Å. The nicotinamide molecules in chains are connected through hydrogen bonds between O and N atoms from the amide moiety. Moreover, the nicotinamide and XN molecules are bonded by the C(6) infinite hydrogen chain in the plane of XN molecules (Figure 6b). The stabilization of the three-dimensional crystal lattice structure is further provided by the hydrogen XN−XN interactions as described earlier. 

#### 2.1.4. XN−GA (1:1)

The XN molecules are packed in the crystal net in the altered layers of AD, BC arrangement (Figure 7a). Glutarimide molecules bind the AD and BC XN layers by the supramolecular hydrogen bonds ring $R_{6}^{6}(16)$ (graph set notation, Figure 7b). The XN molecules are stabilized in the layers by the hydrogen bonds described above. In this structure the π–π stacking interactions play a secondary role in layer binding.

#### 2.1.5. XN−AC (1:1)

The asymmetric unit cell is composed of two XN molecules and one acetamide molecule. The neighboring XN layers are bonded by supramolecular hydrogen bonds between the hydroxyl groups from the XN molecules (B rings) and acetamide (Figure 8). Similar to XN−GA, the $R_{6}^{6}(16)$ supramolecular ring is also formed. In this structure stacking interactions between the rings play a minor role as compared to that of other cocrystals.

#### 2.1.6. XN−CF (1:1)

This cocrystal consists of the XN molecule, caffeine, and two water molecules in the asymmetric part of the unit cell. In this case water is probably coming from atmosphere during crystallization. Similarly to the earlier described, this one consists of layers but both XN and caffeine molecules are in the same plane. The typical pattern of hydrogen bonds between the XN molecules observed for XN−NIC, XN−GA, and XN–AC (Figure 4) is replaced by the net of −XN−caffeine−XN− (Figure 9). The neighboring layers are bonded by the π–π originating from the XN double bond electrons (C8 = C9) and a six-membered caffeine ring (Figure 9). This justifies the crystal tendency to be easily cleaved along the XN−caffeine planes and poor crystal quality (R_1_ = 29%). Water molecules are located in channels along *a* direction and are planar with the layers. As the data quality is very poor, possible hydrogen bonds between the water and the XN or water–caffeine can only be approximated.

### 2.2. Solubility Determination

It has been reported that handling of xanthohumol is difficult since the poor solubility in water (1.3 mg/L at 23 °C [2]) is a major drawback in reproducibility results in the pharmacological studies both in vitro and in vivo [35]. In this way, the application of UV–Vis spectroscopy to compare the solubility of XN with its cocrystals was not possible in pure water, and therefore the solubility experiment was evaluated in a 50:50 (*v*/*v*) ethanol–water solution in accordance to similar cases [36,37]. Due to the relative high solubility of used coformers in aqueous media (several orders of magnitude higher than the value reported for XN), it is suggested that the dissolution profile of XN would be improved after the cocrystals formation. The result of solubility determination of XN and its cocrystalline phases is presented in Figure 10. As expected, all cocrystals investigated exhibit the improvement of solubility, with the largest solubility enhancement demonstrated by the XN−AC cocrystals, which is ca. 2.6 times greater as compared to intact XN. The solubility curve of XN with the maximum solubility value of 7.6 mg/L subsequently reaches a plateau after 30 min. It is shown that the solubility of XN is in the same range as determined by Stevens et al. with 3.5 mg/L (in 5% ethanol) [2]. However, the ‘spring and parachute’ effect described by Guzmán et al. [38], which is commonly observed in case of the enhanced solubility of drug cocrystals did not occur in the study reported herein. Finally, the solubility profiles of pure XN and its cocrystals revealed that the approach of cocrystallization with successful solubility modification was achieved with maximum solubilities of 9.98 (±0.58), 10.54 (±0.98), 12.73 (±0.67), and 20.07 (±0.70) mg/L demonstrated by the XN−CF, XN−NIC, XN−GA, and XN−AC cocrystals, respectively.

### 2.3. Vibrational Spectroscopy

A more profound insight into the XN cocrystal phenomena was obtained through vibrational spectroscopy. The above discussed results are supported by trends in the infrared absorption and Raman spectra. The FTIR and Raman spectra for all of the discussed cocrystals and assignment of specific vibration modes based on the studies carried out for individual components are collected in Appendix A and Appendix A, whereas the spectra of target cocrystals are presented in Figure 11 and Figure 12.

The experimental results show that the FTIR spectra of physical mixtures composed of XN with coformers revealed the linear combination of the bands specific to individual components and no significant shifts were observed, as shown in Appendix A. However, the infrared absorption spectra of the XN cocrystals are definitely differentiated from that of the physical mixture. As follows from Figure 11, there are characteristic changes in positions, relative intensities, and shapes of some vibrational peaks upon cocrystal formation. It has been indicated that in *o*-hydroxychalcones an intramolecular hydrogen bonding between the OH and the carbonyl oxygen atom exists [39]. As discussed above, the O3H3 and O4H4 groups in the XN molecule are involved in the intra- and intermolecular hydrogen bonding so the O1H1 group can undergo hydrogen bonding in the XN cocrystal products easily (as indicated by the presented structures). On the other hand, the above analysis showed that in the structure of XN−CF, all hydroxyl groups from the XN molecule form hydrogen bonds with caffeine. These hydroxyl groups are applied to target the several groups in coformers such as the aromatic nitrogen and the amide groups (acetamide, nicotinamide), the nitrogen and carbonyl groups in the glutarimide ring (glutarimide) and the nitrogen in the imidazole ring (caffeine). 

As a result, in the region 3500−3200 cm^−1^ corresponding to the stretching mode of OH groups, an increase in intensity and broadening of some bands are observed. In particular, the bands at 3440 cm^−1^ and 3202 cm^−1^ (XN−AC), 3389 cm^−1^ and 3192 cm^−1^ (XN−GA), and 3352 cm^−1^ and 3191 cm^−1^ (XN−NIC) were also shifted to lower or higher frequencies as compared to the starting materials. The exception are the XN−CF cocrystals since only a broad band appeared in this region due to water molecules in the structure (Figure 11). A shift from 3160 cm^−1^ in the spectrum of acetamide to 3202 cm^−1^ in the cocrystal spectrum confirms the involvement of the amide group of AC in hydrogen bonding [28]. A new band at 3440 cm^−1^ was related to the presence of the N−H stretching vibrations and the intermolecular N−H⋅⋅⋅O1 interactions with the XN molecule. Similarly, such a spectral pattern was also distinguished in XN−NIC and XN−GA with a new band at 3352 cm^−1^ and 3389 cm^−1^, respectively, and upshift of band at ca. 3190 cm^−1^ in comparison to the pure coformers. 

In the region of 1710–1520 cm^−1^, the FTIR spectra are dominated by the bands attributed to the ring C = C stretching conjugated with the carbonyl group (XN), the C = O (all coformers), the amide I, amide II bands (AC and NIC), and the C = N stretching vibrations (CF and NIC) [40,41,42,43,44]. The intensity of absorption of isolated carbonyls in the XN−CF cocrystals remains almost constant (ca. 1704 cm^−1^), while the band assigned to a conjugated carbonyl group (1660 cm^−1^) is shifted toward lower frequencies as a result of this mode envelopment in hydrogen bonding [43]. The peak at 1681 cm^−1^ due to the –C = O mode of amide moiety of acetamide shifts to 1667 cm^−1^ in the cocrystal demonstrating its involvement in the hydrogen bonding in the cocrystal structure. Interestingly, the C = O amide moiety vibrations of NIC were registered at the same position in the XN–NIC cocrystals indicating that the force constants of the synthons are not strongly changed upon formation of this cocrystal [45]. The intense band in the spectra of all cocrystals centered at 1624 cm^−1^, contributed probably by C = O involved in the intramolecular hydrogen bonding and C = C stretching modes in the XN molecules remains almost unaffected upon cocrystallization. A similar situation refers to two other bands at 1542 and 1513 cm^−1^, assigned to OH bending and C–C ring stretching, except for the XN−CF cocrystal when a broad band centered at 1554 cm^−1^. It is shifted toward higher frequencies as compared to pure caffeine. In the region of the C−OH in-plane bending vibrations a new band appeared at 1391 cm^−1^ in the XN−AC spectrum which was shifted to lower wavenumbers compared to the position in pure acetamide. This band is associated with the vibrational frequencies for the C−N stretch and N−H bends, and the shift of its position confirms the interactions between these groups of XN and acetamide. It is obvious that in the region of bending vibrations of C−OH (phenyl ring), the most significant changes should be revealed for the XN−CF cocrystals due to participation of all hydroxyl groups from XN in the intermolecular interactions. Indeed, the intense band at 1343 cm^−1^ attributed to δOH in XN is downshifted by 5 cm^−1^ in the spectrum of the XN−CF cocrystals. There is also a simultaneous shift of band located at 1291 cm^−1^ in the XN spectrum to 1287 cm^−1^ in the XN−CF cocrystals. 

In general, the strong bands in the FTIR spectrum of a compound correspond to the weak bands in the Raman spectrum. On the other hand, large variations of both the molecular dipole moment and molecular polarizability may be induced by the intramolecular charge transfer between the carbonyl group and phenyl ring through the ethylenic bridge in the XN molecule, then making IR and Raman activity strong simultaneously [46].

The comparison of the Raman spectra of target cocrystals and the starting materials showed the spectra to differ distinctly in the C = C and C = O stretching region. It can be seen from Figure 12 that the vibrational modes of XN (in the form of single crystal) in the spectra range 1700–1520 cm^−1^ have strong bands in both Raman and FTIR spectra at the same time. Except for one exception (XN−CF, Figure 12e), the intense bands at 1628 cm^−1^ and 1610 cm^−1^ are slightly shifted toward lower frequencies in the Raman spectra of cocrystals. Moreover, the ratio of these intensities, to a large extent, remains constant only for the XN−GA cocrystal. Additionally, in the region discussed above, the spectrum of the XN−GA cocrystal has the most broadened bands which is related to the six different arrangements of XN molecules in an asymmetric part of the unit cell. For other cocrystals an increase in the intensity of the band at ca. 1628 cm^−1^ in favor of the second one is observed. This effect is observed in the case of the XN−CF cocrystal, and the downshift of this band is the largest (by 8 cm^−1^) with a parallel increase in the intensities of two bands at 1549 and 1170 cm^−1^. This can be explained by the fact that in the molecular arrangement of XN−CF there are both weak XN interlayer hydrogen bond interactions and π–π stacking. This can be also seen in the region of bending vibrations of the phenyl ring of XN since two intensive bands at about 1470 and 1260 cm^−1^ are not present in the spectrum of XN−CF.

### 2.4. Multivariate Analysis

#### 2.4.1. Principal Component Analysis (PCA)

The restrictions of vibrational spectroscopy due to overlapping bands coming from individual components in the spectra of cocrystals mean that a supplementary tool for cocrystals detection should dispel any doubts. There are various examples of taking advantage of a chemometric approach in the interpretation of the obtained results from the spectroscopy measurements [47,48,49,50]. To gain a more profound insight into the discrimination of physical mixtures with coformers and cocrystals, principal component analysis based on the FTIR and Raman spectra was performed. The contribution of the principal components is shown in Table 2. The first three principal components explain more than 84.5% of the total variance (structure of dependence of primary variables) in all cases. Based on the scree plot (Appendix A) representing the graphic relationship between the eingenvalues and principal components (PCs), three main components were used for further analysis. 

##### Raman Spectroscopy

Figure 13 presents the PCA score plot of PC1 vs. PC2 showing the significant cumulative percentage of total variance (90%). PC1 represents 61.78% of the explained variance and PC2 accounts for 28.06%. These two components distinguish the three groups. The XN−AC and XN−NIC cocrystals and XN create a cluster at the top of the left hand side of the plot and correlate negatively with PC1 and positively with PC2 while XN–CF and XN–GA are separated from others and create two different clusters. The other combinations of the first three PCs (PC1 vs. PC3 and PC2 vs. PC3) are presented in Appendix A. The addition of the third principal component (7.43% of total variance) facilitates discrimination between the cocrystals and the crystal form of XN, especially between the XN–NIC and XN–AC cocrystals. The PC1, PC2, and PC3 loading values for the analyzed region of the Raman spectra are presented in Appendix A. The highest values of the loading plot are located in the 1628–1550 and 1262–1170 cm^−1^ regions. It is worth noting that there is a correlation between the PCA score plot and the loading values [50]. The observations which are positively correlated with PC1 in the score plot have positive peaks on the PC1 loading plot, and vice versa. The peaks positively correlated with the PC1 loading factor describe XN–CF, and those negatively correlated with the PC1 loading factor delineate the other samples, respectively. 

##### FTIR Spectroscopy 

In order to evaluate the main differences in the FTIR spectra, the PCA analysis was made for the two regions: 3700−2700 cm^−1^ and 1800−1000 cm^−1^ for XN, the target cocrystals, and the physical mixtures composed of XN and coformers. In this case the contribution of the total variance to PC1 is 58.53% while PC2 accounts for a lower percentage of the total variance (26%). The PCA score plots for the first two PCs are shown in Figure 14a. PC1 correlates positively with the XN, XN−CF, XN−GA, XN−NIC, and XN−CF_mix_ samples, but negatively with the XN−AC cocrystals, and physical mixtures (XN−AC_mix_, XN−GA_mix_, and XN−NIC_mix_). The XN−NIC, XN−GA, and XN−CF cocrystals are grouped closely to each other and form a cluster in the central part of the PCA score plot, except for XN–AC. Probably a new band ~3440 cm^−1^ in the spectrum of XN–AC can determine this separation. Noticeably, physical mixtures are located around cocrystals and are further separated from the center of the score plot. In the high wavenumber range, the XN sample shows insignificant similarities to the XN–CF cocrystals. After the addition of the PC3 component (Appendix A) which accounts for 10.32% of the total variance, it can be observed that the cocrystals and physical mixtures are distinctly separated and located in different positions. The loading plots for PC1, PC2, and PC3 are presented in Appendix A. This shows that the region between 3443 and 3180 cm^−1^ has a significant impact on the evaluation of similarities and differences between the samples.

In the fingerprint region, PC1 represents 52.05% and PC2 21.37% of the total variance. The PCA score plots for the studied samples are shown in Figure 14b. It is observed that the XN, XN−NIC, XN−AC, XN−NIC_mix_, and XN−AC_mix_ samples are positively correlated with PC1 whereas XN−CF, XN−GA, XN−CF_mix_, and XN−GA_mix_ correlate negatively. Noticeably, in this spectral range PC2 is helpful to differentiate cocrystals (positively correlated) in physical mixtures (negatively correlated). However, the XN−AC cocrystal is located in closer proximity to physical mixtures than other cocrystal samples. Yet most importantly, XN is clearly separated from other samples. As expected, the first two PCs allow cocrystals to be distinguished from pure XN and physical mixtures. The score plots for PC1 vs. PC3 and PC2 vs. PC3 are shown in Appendix A. Loadings factors for the first three PCs are presented in Appendix A. The highest values of loadings are observed for the region of 1705 – 1553 cm^−1^ which are attributed to the C = C, C = O, and C = N stretching vibrations.

#### 2.4.2. Hierarchical Clustering Analysis (HCA)

Figure 15 presents the tree diagrams obtained from the hierarchical clustering analysis using the data acquired from the Raman and FTIR spectra in the same spectral range as the PCA analysis. The use of the complete linkage and average linkage methods of agglomeration gives similar results for the data presented in the Raman spectra (Figure 15a,b). The results are cut off at about 3 units of distance and form three clusters. The XN, XN−NIC, and XN−AC cocrystals create the first cluster (2.4 unit of distance) and display the largest similarity, whereas the XN−CF and XN−GA samples create two subsequent clusters (Figure 15a). This is similar to those found in the PCA score plot (Figure 13). In the other method the results are cut off at about 0.2 units of distance (Figure 15b). In this case, the first cluster is gradually formed by the XN, XN−NIC, XN−GA, and XN−AC cocrystals. Interestingly, the XN−CF sample is grouped in a separate cluster in both methods. 

As shown in Figure 15c,d, HCA distributes the cocrystals and physical mixtures into four clusters. The XN and XN−CF samples are grouped in the first cluster at the shortest distance which corresponds to the results from the PCA score plot presented in Figure 14a. The XN−NIC_mix_ sample creates a separate cluster which is the most distinct from the others. It should be noticed that the cocrystals and physical mixtures composed of the same conformer form individual clusters, except for the XN−NIC and XN−NIC_mix_ samples. Taking a high wavenumber region into consideration, the samples with nicotinamide show the most prominent differences (Figure 15c,d).

In the case of the fingerprint region, two different methods of measuring the distance (Euclidean and Chebyshev) were used. The results presented in Figure 15e are grouped into four clusters. The XN−NIC and XN−NIC_mix_ samples show the smallest similarities as compared to the other samples (cocrystals and physical mixtures with the same coformer) which are consistent with the above mentioned observations in the region 3700−2700 cm^−1^. On the other hand, the tree diagram created using the Chebyshev distance in Figure 15f reveals that the XN sample forms an individual cluster whereas the physical mixtures and suitable cocrystals are merged. This is similar to that calculated by PCA analysis (Figure 14b). Basically, HCA can be used directly with the PCA calculations to simplify the method of discrimination between the cocrystals and physical mixtures.

## 3. Materials and Method

### 3.1. Materials

Xanthohumol purchased from Sigma-Aldrich (Darmstadt, Germany) with purity of >98% was recrystallized from acetonitrile. Nicotinamide, glutarimide, acetamide, caffeine, and all used solvents were of analytical grade and were purchased from Sigma-Aldrich. The compounds used as coformers are included in the list of compounds authorized for the safe use as additives to food products, pharmaceutical preparations, and dietary supplements (so-called GRAS list (“Generally Recognized as Safe”) created by the FDA-American Food and Drug Agency).

Synthesis of the cocrystals. Xanthohumol and coformers at the 1:1 molar ratio were dissolved in acetonitrile at 30 °C. The concentrated solution was kept for 2 days at room temperature to obtain yellow crystals (with a plate and prismatic shapes) using the slow evaporation solution growth technique. Then crystals were collected from the crystallization vessels.

### 3.2. Methods

#### 3.2.1. Single Crystal X-ray Diffraction 

Single crystal diffraction data were collected on a Rigaku Oxford Diffraction diffractometer equipped with a MicroMax-007 HF, Cu rotated anode as an X-ray source (CuKα), multilayer optic, and a Pilatus 300 K area detector at T = 293 K. The 2θ was measured in a range of 6–110° with a resolution of 0.078° and 10 min counting time per frame. Data reduction and cell refinement were carried out using CRYSALIS^PRO^ [51]. All structures were solved applying direct methods [52] and refined using the Olex2 program [53]. The refinement was based on the squared structure factors (F^2^) for all reflections except those with very negative F^2^ values. Almost all hydrogen atoms were located in the idealized averaged geometrical positions. The scattering factors were taken from Tables 6.1.1.4 and 4.2.4.2 in [54]. Table 1 includes the experimental details for single crystals measurements. The presented structures were deposited in the Cambridge Crystallographic Data Centre (CCDC) with the numbers: XN−AC 1,955,279, XN−GA 1,955,276, XN−NIC 1,955,269, and XN−CF 1,955,267. 

#### 3.2.2. X-ray Powder Diffraction (PXRD)

The PXRD patterns of all cocrystals were obtained using the same diffractometer as for the single crystal analysis (Rigaku, Tokyo, Japan) but working in a powder diffraction mode. The samples were mounted in the borosilicate glass capillary tubes (wall thickness of 0.01 mm and diameter of 1 mm). The measured range was from 6.4−50°. For averaging, the sample was rotated around the phi axis. The data were collected using the CRYSALIS^PRO^ software [51]. New identified phases were selected for further studies. Some results are presented in Appendix A.

#### 3.2.3. Hirshfeld Surface Analysis

The Hirshfeld surface analysis and fingerprint plots were generated by the CrystalExplorer 17.5 program. The plots were used to describe various intermolecular interactions especially H-bonds which are the most important in the stabilization of crystal net and other contacts present in the structure of XN−NIC, XN−CF, XN−AC, and XN−GA cocrystals. The crystallographic information file (CIF) was used as input for the analysis. Directions and strengths of intermolecular interactions within the crystals were mapped onto the Hirshfeld surfaces according to the description reported by Abidi et al. [55]. The results of the Hirshfeld surface analysis were presented in Appendix A. 

#### 3.2.4. Solubility Experiment

Solubility of pure XN and its cocrystals was determined according to the method employed by Sowa et al. [36] with minor modifications of the amounts of ingredients. Shortly, 1 mg of powder XN and cocrystals were suspended in 5 mL of 50:50 (*v*/*v*) ethanol–water mixture and stirred in a thermostated vessel at 27 °C. An aliquot of the samples was transferred from the slurry at the intervals of 5, 10, 15, 20, 25, 30, 35, 45, 50, 55, 60, and 90 min, then filtered with a 0.45 μm nylon filter. The electronic absorption spectra were recorded with a double-beam UV−Vis spectrophotometer Cary 300 Bio (Varian) equipped with a Cary Peltier temperature control. The samples were measured in closed quartz cuvettes (Helma) with a pathlength of 1.0 cm in the wavelength range of 200−600 nm. The absorption coefficient of XN was determined (ε = 24,319 dm^3^/mol⋅cm) from the slope of the absorbance measured at 373 nm vs. the XN concentration in a 50% aqueous EtOH mixture. 

#### 3.2.5. FTIR Spectroscopy

The Fourier-transformed infrared absorption spectra were recorded with a 670-IR Varian spectrometer. Typically, 16 scans were collected at the resolution of 4 cm^−1^. Spectra were obtained in the region between 4000 and 600 cm^−1^. The laser power was set to 50% (0.25 mW). The measurement was made in pellets of KBr previously dried (1 day) at the temperature of 120 °C. The FTIR spectra were obtained for all starting materials (XN, coformers), the physical mixture of them (for comparison), and cocrystals. The spectral analysis was processed with the use of the Grams/AI 8.0 software (Thermo Scientific, Waltham, MA, USA).

#### 3.2.6. Raman Spectroscopy

All Raman spectra were recorded with a Raman microscope inVia Reflex System from Renishaw (Renishaw Plc., Wotton-Under-Edge, UK), equipped with a charge-coupled device (CCD) detector with a resolution of 1 cm^−1^. Spectra were collected in triplicate using a 50/0.75 × NA objective, with accumulation of 1 scan and 10 s exposure time. The excitation was provided by the 514 nm line of an argon ion laser with the power set as 1% or 10% from 5 mW (in this case the best spectra were selected). In every case the laser beam was focused on a surface of single crystal placed on an aluminum support. The spectra analysis and correction of a background signal originating from the fluorescence were performed using the Grams/AI 8.0 software (Thermo Scientific, Waltham, MA, USA). All Raman spectra were normalized to the intensity of the most intense band in the range 1750–600 cm^−1^.

#### 3.2.7. Multivariate Analysis Principal Component Analysis (PCA) 

##### Principal Component Analysis (PCA)

The Principal component analysis is one of the exploratory methods used to detect data structure by searching the explanation of the correlation structure of a set of variables using a smaller set of linear combinations of these variables. The aim of PCA is data reduction with creation of a new set of uncorrelated variables called principal components (PCs) which will explain the largest possible variances [56,57]. The general PCA model can be expressed: X = TP^T^(1)
where the X matrix is decomposed of the dataset into two smaller matrices T and P (the scores matrix and loadings matrix, respectively). The loadings illustrate the weight or importance of each variable within the original data and represent the variance for each variable (in this case the wavenumber). Loadings of the PCs are a significant tool for interpretation of the source of the variability inside a dataset from the spectra.

##### Hierarchical Clustering Analysis (HCA) 

Another chemometrics method that can be a continuation of PCA classification is the hierarchical clustering analysis. HCA is an unsupervised method in which the focus is on the classification problem. The aim of the HCA is to detect similarities in the variables set and classification into clusters. Different classification algorithms can be applied in this method. 

Principal component analysis (PCA) and hierarchical cluster analysis (HCA) were used to obtain Raman and FTIR spectra for the cocrystals and crystal form of xanthohumol. Before the data analysis, all spectra were subjected to pre-treatment (multi-point baseline correction, Savitzky-Golay smoothing with 11 points and Y offset correlation, points were set to zero) using the Grams/AI 8.0 software (Thermo Scientific, Waltham, MA, USA). The analysis was conducted in the range 1700–1000 cm^−1^ for the Raman spectra and 3700–2700 cm^−1^ and 1800–1000 cm^−1^ for the FTIR spectra. In the principal component analysis the covariance matrices were used. In the hierarchical cluster analysis the Euclidean, Chebyshev, and Pearson correlation distance between the pairs of samples were used as a distance measure. Complete and average linkage criteria were used as an agglomeration method. The multivariate analysis was prepared using the Statistica 13 software (TIBCO Software Inc. Palo Alto, CA, USA). 

## 4. Conclusions

In summary, for the first time the synthesis of new xanthohumol cocrystals with pharmaceutically acceptable coformers, such as nicotinamide, glutarimide, acetamide, and caffeine in the 1:1 stoichiometry obtained by the slow evaporation solution growth technique was reported. The chemical characteristics of the cocrystals are confirmed by single crystal X-ray diffraction, FTIR, and Raman techniques. The analysis of packing and interactions in the crystal lattice revealed that molecules in the target cocrystals were packed into almost flat layers, formed by the O–H⋅⋅⋅O, O–H⋅⋅⋅N, and N–H⋅⋅⋅O-type contacts between the xanthohumol and coformer molecules. The single crystal X-ray diffraction study indicated the existence of characteristic hydrogen bond motif in the structures of XN−NIC, XN−GA, and XN−AC cocrystals resulting from the involvement of the hydroxyl group (O4H4) and carbonyl oxygen (O2) from the neighboring XN molecule in layers stabilization. Except for XN−CF, the hydrogen bonds from the hydroxyl group (A ring) were the main interlayer binding synthons. In the XN cocrystals with caffeine, the neighboring layers were weakly bonded by both the hydrogen bonds and the π–π stacking which could account for the low quality of the XN−CF cocrystals. The XN−GA and XN−AC cocrystal structures were sustained by the supramolecular hydrogen bonds ring $R_{6}^{6}(16)$, while the XN−NIC cocrystals were bonded by the C(6) infinite hydrogen chain in the plane of XN molecules. Furthermore, in all structures under study, the π–π stacking interactions additionally stabilized layer binding. FTIR spectroscopy confirmed that the crystalline phase of the cocrystals was not only a physical mixture of the initial reactants. A completely different crystal phase reflected the spectral changes, mainly due to the formation of hydrogen bonding between the XN and coformers molecules. However, the application of Raman spectroscopy gave the insight, to a lesser extent, into the nature of the interactions in the cocrystals.

It was shown that the combination of multivariate statistical methods (PCA and HCA) with the Raman and FTIR spectroscopic data is a screening tool employed for identifying new cocrystal structures and could accelerate this process significantly. Finally, the solubility studies of XN cocrystals revealed a 2.6-fold increase in the XN solubility for XN−AC and, to a lesser extent, for XN−GA, XN−NIC, and XN−CF (ca. 1.6-, 1.4- and 1.3-fold, respectively) as compared to intact XN. Based on the numerous pharmacological activities of XN, the results of the above investigations suggest that the cocrystals can be prospective candidates for future clinical applications. Moreover, determination of the ability to penetrate biological membranes by the new molecular forms of XN in comparison with pure compounds will be the subject of the next paper.

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
