# Peer review of "Formation of Prenylated Chalcone Xanthohumol Cocrystals: Single Crystal X-ray Diffraction, Vibrational Spectroscopic Study Coupled with Multivariate Analysis"

_molecules, 2019, doi:10.3390/molecules24234245_

Round 1

Reviewer 1 Report

The submitted study presents four cocrystals of xanthohumol with nicotinamide, glutarimide, acetamide, and caffeine as coformers. Single crystals of the coformers were prepared through slow solvent evaporation. With the exception of xanthohumol-caffeine cocrystal, high quality single crystals were obtained and single crystal X-ray diffraction unambiguously identified the molecular packing and interactions within the cocrystals. While the crystallographic part of the manuscript is clear and well presented, there are two main issues to be resolved before reconsideration.

1) “Bioavailability of orally administrated xanthohumol is considered to be extremely low” (line 51); “improvement of the XN bioavailability… modification of physicochemical properties…” (line 54-55); “cocrystallization… to modifiy water solublity, dissolution rate…” (line 56-57)

This part of introduction implies that the problem related to the bioavailability of xanthohumol can be solved through the cocrystallization of xanthohumol. Yet no physicochemical properties of the cocrystals were presented in this study. While the authors suggested that studies on the solubility and membrane permeation will be dealt in the next paper (at the end of the Conclusions section), this reviewer feels that some results on the improvement should be included in the present paper to justify the cocrystal approach.

2) On the same note as point 1, it has to be clarified if the improvement of solubility and permeability would solve the problem of bioavailability when “rapid excretion or extensive metabolism (line 52)” are the underlying reason. Also, in vivo cyclization of the chalcone to the flavonoid was suggested in reference 19.

3) Section 3 (Multivariate Analysis) should be substantially reduced to really focus on the fast screening of new cocrystal phases. The current description is rather confusing, and it is unclear if the approach can be really utilized to identify and categorize new cocrystals as the first screening tool. In addition, this part per se looks inessential for this study after single crystal X-ray diffraction. 

Minor points:

1) Dr. Kaminsk’s contact information as the co-corresponding author?

2) “B ring” in the Abstract can be more specific for the readers unfamiliar to the xanthohumol structure.

3) Figure 1: Usual chemical structure in addition to the current figure would be helpful. Also, maybe indicate A ring and B ring in the figure?

4) Figure 2: Scale bars are hard to decipher especially from b to e.

5) Figure 3a: Change to the same orientation (top view) as b, c, and d?

6) line 127: spherical hindrance? Steric hinderance?

7) line 144: any explanation on the 2.122 Angstrom interaction?

8) line 178: need to explain the inclusion of water molecules when the solvent was acetonitrile. Perhaps, ambient humidity?

9) Consider to move “The Hirshfeld surface” to the supporting information.

Author Response

Response to Reviewer #1

First of all, we would like to express our deepest gratitude to the Reviewer for the valuable guidance which undoubtedly have enriched this manuscript. We agree with most of comments. We addressed each of them in the revised manuscript.

Major points:

Point 1: “Bioavailability of orally administrated xanthohumol is considered to be extremely low” (line 51); “improvement of the XN bioavailability… modification of physicochemical properties…” (line 54-55); “cocrystallization… to modifiy water solublity, dissolution rate…” (line 56-57).

This part of introduction implies that the problem related to the bioavailability of xanthohumol can be solved through the cocrystallization of xanthohumol. Yet no physicochemical properties of the cocrystals were presented in this study. While the authors suggested that studies on the solubility and membrane permeation will be dealt in the next paper (at the end of the Conclusions section), this reviewer feels that some results on the improvement should be included in the present paper to justify the cocrystal approach.

Response 1: We would like to thank the Reviewer for drawing attention to weak points. Firstly, regarding the improvement of the XN bioavailability through the cocrystallization process which in the opinion of Reviewer was an insufficient discuss, we agree to some points. In Introduction section we decided to give a picture of current problem with XN bioavailability. It was a background of primary objective, which concerned the synthesis of four novel XN cocrystals with pharmaceutically acceptable coformers. However, we agree that the solubility determination is the key experiment in cocrystal development. In the new version of manuscript we included the results from the solubility studies of pure XN and its cocrystals (see the solubility curves presented in Figure 10, and a description of method in the P.3.2.4. Solubility experiment).

Point 2:  On the same note as point 1, it has to be clarified if the improvement of solubility and permeability would solve the problem of bioavailability when “rapid excretion or extensive metabolism (line 52)” are the underlying reason. Also, in vivo cyclization of the chalcone to the flavonoid was suggested in reference 19.

Response 2:

According to the next part of Reviewer`s comments, chalcones, similar to flavonoids, generally exhibit low oral bioavailability due to their poor aqueous solubility. To address this problem, various strategies have been developed to improve oral absorption of poorly water-soluble compounds such as cocrystals, amorphous solid dispersions, nanotechnology and lipid formulation [Zhao J. et al. Improvement strategies for the oral bioavailability of poorly water-soluble flavonoids: An overview. Int. J. Pharm. 2019; 570:118642; doi: 10.1016/j.ijpharm.2019.118642]. Moreover, the solubility and permeability properties are the major factors used to describe oral absorption according to the biopharmaceutics classification system (BCS) [Amidon G.L. et al. A theoretical basis for a biopharmaceutic drug classification: the correlation of in vitro drug product dissolution and in vivo bioavailability. Pharm. Res. 1995; 12: 413-20. doi:10.1208/s12248-014-9620-9]. Numerous studies have been reported that cocrystals can modify properties of active compounds without any impact on their pharmacological activity, moreover, also optimize their physicochemical properties such as stability, solubility, permeability and bioavailability [Sathisaran I. et al. Engineering Cocrystals of Poorly Water-Soluble Drugs to Enhance Dissolution in Aqueous Medium, Pharmaceutics. 2018 31;10(3); doi: 10.3390/pharmaceutics10030108]. It was suggested that the mechanisms which are responsible for the improved cocrystal solubility are related to the changed lattice and solvation energies by the presence of the coformer [Babu N. et al. Solubility advantage of amorphous drugs and pharmaceutical cocrystals. Cryst Growth Des 2011; 11: 2662-79. doi:10.1021/cg200492w].

Although the bioavailability of orally administrated xanthohumol is predicted to be extremely low, considerable concentrations of XN might be reached in the cytosol. Binding to cytosolic proteins via electrophilic sites, e.g. the a-unsaturated carbonyl group also strongly influences the distribution of XN [Pang Y. et al.; Binding of the hop (Humulus lupulus L.) chalcone xanthohumol to cytosolic proteins in Caco-2 intestinal epithelial cells. Mol. Nutr. Food Res. 2007, 51, 872-9]. The mechanism of covalent binding of XN with specific proteins, could be responsible for significant XN biological activities such as cancer prevention [Yoshimaru T. et al.; Xanthohumol suppresses oestrogen-signalling in breast cancer through the inhibition of BIG3-PHB2 interactions. Sci. Rep. 2014, 4, 7355] In this context, the prediction of the bioavailability of XN molecule, as well as its specific interactions with enzymes depends to a great extent on its solubility in aqueous solutions. It should be also reported that the cellular experiments performed with XN have been troubled by the low solubility of XN in cell culture medium. Moreover, in order for any active compounds to be transported through the membranes, the molecule must be dissolvable in the aqueous phase of the intestinal fluid. Even if there is extensive metabolism of XN in human body, it doesn’t mean that the total amount of XN was converted to its metabolites.

Point 3: Section 3 (Multivariate Analysis) should be substantially reduced to really focus on the fast screening of new cocrystal phases. The current description is rather confusing, and it is unclear if the approach can be really utilized to identify and categorize new cocrystals as the first screening tool. In addition, this part per se looks inessential for this study after single crystal X-ray diffraction.
Response 3: As suggested by the reviewer, in the revised manuscript, we reduced this part of section. Figure 14 and 16 was moved to the Supplementary Materials and now are presented as Figure S8 and Figure S11.

Minor points:

Point 1: Dr. Kaminski’s contact information as the co-corresponding author?

Response 1: It has been added according to Reviewer's suggestion.

Point 2:  “B ring” in the Abstract can be more specific for the readers unfamiliar to the xanthohumol structure.

Response 2: We decided not to make changes regarding this part of abstract because the structure of XN is similar to the structure of well-known flavonoids which also consisted two phenyl rings (named A and B).  

Point 3: Figure 1: Usual chemical structure in addition to the current figure would be helpful. Also, maybe indicate A ring and B ring in the figure?

Response 3: Necessary changes have been made and a new Figure 1a should be acceptable.

Point 4: Figure 2: Scale bars are hard to decipher especially from b to e.

Response 4: The scale bars used in Figure 2b-e have been changed according to Reviewer's suggestion.

Point 5: Figure 3a: Change to the same orientation (top view) as b, c, and d?

Response 5: Necessary changes have been made and a new Figure 3a should be acceptable.

Point 6:  line 127: spherical hindrance? Steric hinderance?

Response 6: It was corrected.

Point 7:  line 144: any explanation on the 2.122 Angstrom interaction?

Response 7: Necessary changes have been made (lines 151-153).

Point 8:  line 178: need to explain the inclusion of water molecules when the solvent was acetonitrile. Perhaps, ambient humidity?

Response 8: It was explained (lines 192-193)

Point 9: Consider to move “The Hirshfeld surface” to the supporting information.

Response 9: As suggested by the Reviewer, in the revised manuscript, we have reduced the part of text regarding “The Hirshfeld surface” and moved to the Supporting Materials (Figures S1-S3).

Reviewer 2 Report

This is a nice MS by Budziak et al., wherein the synthesis of xanthohumol cocrystals with pharmaceutically acceptable coformers, such as nicotinamide, glutarimide, acetamide and caffeine is reported. A suite of techniques including XRD, FT-IR and Raman were employed to characterize of the cocrystals. Overall, the MS is concise, nicely written and experimental data supported the claims. Given that, this reviewer is happy to recommend its publication in Molecules.

Author Response

Thank you

Reviewer 3 Report

p.1.

too many keywords

p.5.

Authors state:
"In the case of XNNIC due to the center of symmetry the XN molecule forms a synthon with carbonyl oxygen from nicotinamide molecule"

What is the connection of the center of symmetry existence with the structure of synthon? Center of symmetry can be or not present in the molecule or more complex structure, but it has nothing to do with the structure of the synthon. It can be mentioned as a structural feature, but it can not be the reason for forming structure of synthon.

p.11.

"the principal component analysis based on the FT-IR and Raman spectra was developed"

PCA can be "performed" but it was "developed" several decades ago.

p.12.

Spectral area for PCA were different for Raman and for IR. What is the reason for that?

In addition, IR spectra were divided into two region and SEPARATE PCA were performed on each region. Authors should clearly explained why did they separate these region and reasons for that.

p. 17 and supplementary

authors state that "All Raman spectra were normalized at the maximum intensity in the range 1750  600 cm−1."

It is not clear whether the spectra were normalized based on the Raman intensity at one particular wavenumber or by integrated intensity of whole band. This should be clarified.

p. 18 in 4. Conclusions

in the sentence:

"Furthermore, in all structures under study, the π-π stacking interactions played a secondary role in a layer binding."

it is not convincing how the authors made this conclusion. pi-pi stacking interactions are usually important in layer binding and additional explanation should be provided.

Author Response

Response to Reviewer #3

Thank you for the valuable comments, we appreciate your time and the effort you have made to review our paper. The article was corrected according to the comments provided in the review and all the statements were addressed below.

Point 1. too many keywords

Response 1: It has been changed according to Reviewer's suggestion (lines 33-35).

Point 2. Authors state: "In the case of XN-NIC due to the center of symmetry the XN molecule forms a synthon with carbonyl oxygen from nicotinamide molecule". What is the connection of the center of symmetry existence with the structure of synthon? Center of symmetry can be or not present in the molecule or more complex structure, but it has nothing to do with the structure of the synthon. It can be mentioned as a structural feature, but it cannot be the reason for forming structure of synthon.

Response 2: You are right, therefore, we removed this state.

Point 3. p.11. "the principal component analysis based on the FT-IR and Raman spectra was developed". PCA can be "performed" but it was "developed" several decades ago.

Response 3: You are right. It was an unfortunate oversight. The sentence has been corrected.

Point 4. p.12. Spectral area for PCA were different for Raman and for IR. What is the reason for that?

In addition, IR spectra were divided into two region and SEPARATE. PCA were performed on each region. Authors should clearly explained why did they separate these region and reasons for that.

Response 4: FTIR and Raman spectroscopy are complementary techniques, and FTIR spectra are due to polar functional groups while the bands in Raman spectra are due to nonpolar functional groups. In general the strong bands in the FT-IR spectrum of a compound correspond to the weak bands in the Raman spectrum. Concerning the question about the differences in the spectral for the PCA analysis, since no intense bands were observed for the region of 3700 – 2700 cm−1 in Raman spectra of the target cocrystals, data evaluation was limited to the fingerprint region (1700–1000 cm−1). Moreover, the region left out of the analysis presented an overall low signal-to-noise ratio and could induce misclassification. According to the next part of Reviewer`s comments, PCA was performed in two spectral regions in order to carry out the analysis in the same wavenumber regions both for Raman and FT-IR spectroscopy. As mentioned above, we had to excluded the high wavenumber region from further Raman analysis. Additionally, each of this region is attributed to the vibrational modes related to different functional groups involved in the interactions. The spectral region of 3700 – 2700 cm−1 contains very limited spectral information whereas the bands between 1700 and 1000 cm−1reflect the most important structural evidences, especially the involvement of the carbonyl groups, amide and C=N groups in the intramolecular hydrogen bonding.

Point 5. p. 17 and supplementary. Authors state that "All Raman spectra were normalized at the maximum intensity in the range 1750 - 600 cm−1." It is not clear whether the spectra were normalized based on the Raman intensity at one particular wavenumber or by integrated intensity of whole band. This should be clarified.

Response 5: Thank you for the comment. It has been changed according to Reviewer's suggestion. The sentence now reads: “All Raman spectra were normalized to the intensity of the most intense band in the range 1750 - 600 cm−1” (line 509).

Point 6. p. 18 in 4. Conclusions. In the sentence: "Furthermore, in all structures under study, the π-π stacking interactions played a secondary role in a layer binding." It is not convincing how the authors made this conclusion. pi-pi stacking interactions are usually important in layer binding and additional explanation should be provided.

Response 6:

We agree, because we did not have any calculations to prove the interaction energy, therefore we cannot say which the interaction is stronger and more important (pi-pi or H-bond). Necessary changes have been made (line 25 and lines 556-557).

Round 2

Reviewer 1 Report

The authors added the solubility study in the revised manuscript. This greatly improved the overall form of the presented research. This reviewer does not have any more issues to discuss.